# MFF-YOLO: An Accurate Model for Detecting Tunnel Defects Based on Multi-Scale Feature Fusion

**DOI:** 10.3390/s23146490

**Published:** 2023-07-18

**Authors:** Anfu Zhu, Bin Wang, Jiaxiao Xie, Congxiao Ma

**Affiliations:** School of Electronic Engineering, North China University of Water Resources and Electric Power, Zhengzhou 450045, China; z20211181076@stu.ncwu.edu.cn (B.W.); z202210181146@stu.ncwu.edu.cn (J.X.); z20201181029@stu.ncwu.edu.cn (C.M.)

**Keywords:** deep learning, target detection, multiscale, feature fusion

## Abstract

Tunnel linings require routine inspection as they have a big impact on a tunnel’s safety and longevity. In this study, the convolutional neural network was utilized to develop the MFF-YOLO model. To improve feature learning efficiency, a multi-scale feature fusion network was constructed within the neck network. Additionally, a reweighted screening method was devised at the prediction stage to address the problem of duplicate detection frames. Moreover, the loss function was adjusted to maximize the effectiveness of model training and improve its overall performance. The results show that the model has a recall and accuracy that are 7.1% and 6.0% greater than those of the YOLOv5 model, reaching 89.5% and 89.4%, respectively, as well as the ability to reliably identify targets that the previous model error detection and miss detection. The MFF-YOLO model improves tunnel lining detection performance generally.

## 1. Introduction

With China’s recent economic growth, the tunnel sector has also entered a golden age of development, and tunnel building has emerged as one of the crucial elements of China’s infrastructure development. However, due to geological factors, poor construction practices, and natural calamities, tunnels may have concealed flaws including uncompacted, hollow, and water filling that gravely jeopardize their service life [1].

To identify and address issues with the tunnels as soon as they arise, regular inspection and maintenance are required. However, conventional methods for finding tunnel defects, such as visual inspection and acoustic inspection, have low detection efficiencies and high result error. Given the quick advancement of deep learning technologies in the detection of targets, such as damage detection and localization of bridge deck pavement [2], crack detection in concrete bridges [3,4], steel surface flaw detection [5,6], and wheel defect detection [7] across a variety of disciplines in society, science, and engineering, deep learning-based tunnel flaw detection has recently gained the attention of both domestic and international academics. For example, Sjölander et al. [8] summarized the research on the application of optical detection technology and autonomous evaluation methods based on machine learning technology in tunnel lining inspection, as well as the research on digital cameras, laser scanning, fiber optic sensors, and other methods; they also proposed issues with traditional tunnel inspection methods, such as their low efficiency. Deep learning technology was used by Maeda et al. [9] to address the issue of data acquisition and traditional data enhancement techniques like rotation, translation, and flip that may alter the semantic information of the image and cause other drawbacks, using methods like selective image cropping and stitching to address the issue of the insufficient data set; however, the model detection effect did not improve. With the primary objective of resolving the data signal-to-noise ratio and multi-path interference problems to offer assurances for data feature extraction, Lei et al. [10] suggested an air-coupled geo-radar detection technique provided by F-K filtering and BP migration. A strong technical guarantee for single defect detection was provided by the GPR forward simulation model developed by Wu et al. [11], but since tunnel defects are frequently made up of hollow, water-filling, and non-compactness, the model’s generalization ability is not ideal. Ali et al. [12], using Faster-RCNN and YOLOv3 networks as well as conventional detection techniques, examined the performance of concrete structures; the findings revealed that the convolutional neural network-based detection strategy had greater detection accuracy and localization precision. Before training the model with an enhanced U-Net network, Wang et al. [13] first performed picture preprocessing. Smaller and more subtle faults might be detected by the model, but the model’s efficiency was low, and its real-time performance could not keep up with demand. Li et al. [14] used U-Net and alternating update convolutional neural networks for automatic tunnel defect detection: first, the image is segmented and predicted to extract defect features; next, alternating update convolutional neural networks are used for classification and localization; however, the model is more complex and cannot satisfy the tunnel defect detection engineering requirements.

Although several researchers have developed numerous techniques for finding tunnel defects, these techniques still have flaws and disadvantages, such as low detection accuracy and slow detection speeds. Therefore, we are working on the relevant research and attempting to address the current shortcomings to further enhance the effectiveness and accuracy of tunnel defect detection. To increase the accuracy and usefulness of tunnel defect detection, as well as to better support and guarantee tunnel construction and maintenance operations, we will keep researching new methods and approaches. Previous studies by our group have included improvements to SGD networks and residual modules [15], the introduction of adversarial networks for data expansion problems [16], and the use of neural network fusion techniques to improve the generalization performance of the model.

This study addresses the current issues with tunnel defect detection in two ways: first, in terms of data acquisition, the most recent radar detection equipment’s working process is shown in Figure 1, which has the qualities of high accuracy and high resolution and can precisely identify the defects within the depth of 10 m underground through the on-site survey; second, in terms of data processing, a detection model based on multi-scale feature fusion technology is proposed in this study, which uses the MFF-YOLO method to detect defects.

The following are this paper’s key contributions:

(1) Designing the Weighted Cross Connections Feature Pyramid Network (WCFPN) to address the issue of missing feature map information.

(2) Designing the Re-weighted Non-Maximum Suppression (RWNMS) to address the issue of redundancy and miss detection of detection frames.

(3) Improving the loss function by integrating the aspect ratio and Euclidean distance factors to improve the model training efficiency and detection accuracy.

The remainder of the text is organized as follows: Section 2 discusses the work on multiscale feature fusion and convolutional neural networks; Section 3 describes the technological strategies for improvement; the data set and assessment metrics are introduced in Section 4; tests are carried out in Section 5 to confirm the improved model performance; and the work is reviewed and expected in Section 6.

## 2. Related Work

### 2.1. Convolutional Neural Network

CNN [17] is a deep learning model consisting of convolutional, pooling, and fully connected layers, etc., for implementing tasks such as image classification, target detection, and image segmentation. One of the target detection methods can be divided into two-stage and single-stage, and the difference between the two algorithms lies in the way the generation frame is combined with the candidate frame. In a two-stage algorithm, candidate regions are first generated and then the CNN applies to these candidate regions for classification. For example, the R-CNN [18,19,20,21,22] family of algorithms has high detection accuracy but is computationally intensive and inefficient. In contrast, single-stage algorithms such as the SSD [23,24,25,26,27] and YOLO [28,29,30,31,32,33,34,35] series can achieve significant detection speedups. Single-stage target detection algorithms have become preferred in industrial applications due to their ability to directly output information about the position and detection frame of the target to be detected. In conclusion, with the continuous development of deep learning techniques, target detection algorithms are also constantly being advanced and optimized, with different algorithms being suitable for different application scenarios and needs.

### 2.2. Multi-Scale Feature Fusion

Multi-scale feature fusion is a technique for combining feature maps at different scales to improve the performance of computer vision tasks. Common approaches include cascade structures [36,37,38], pyramid networks [39,40,41,42] and attention mechanisms [43,44,45]. Cascade structures link feature maps at different scales together to form cascade networks, which can be effective in improving performance; pyramid networks are a hierarchical approach to image processing that extracts features at different scales and combines them; and attention mechanisms can make the network focus more on important features by weighting feature maps at different scales. All these methods aim to improve the performance of computer vision tasks by using feature maps at different scales, thus improving the accuracy and robustness of the task.

## 3. Methods

The MFF-YOLO model improvement approach is described in this section, and Figure 2 illustrates the structural characteristics of the MFF-YOLO model. The model improvement consists of two main components: the WCFPN structure of the neck network, where the blue module represents the Weighted Cross Connections built by weighting and the red line portion reflects the network across connection idea; and the RWNMS mechanism at the predicted end, where the EIOU loss function is used in the screening process of the prediction frame and the weights are further provided by normalization.

### 3.1. WCFPN

The target detection model is prone to the problem that as the number of layers in the feature map increases, the resolution of the feature map decreases, resulting in poor information transfer in the feature map and hence the lack of detection capability of the model. The model may zoom in and out of the feature maps by adding operations like convolutional layers and pooling layers; however, this reduces the performance of the model since they commonly cause information loss or information redundancy. The study uses cross-scale linking to connect feature maps of multiple sizes, better using the data in those feature maps, and then suggests the concept of weighted fusion when zooming in and out on the feature maps to address this issue.

Using the effective feature fusion technique known as weighted fusion, data are combined in feature maps of various scales according to predetermined weights to create multi-resolution feature maps. Cross-scale linking, unlike weighted fusion, is a feature alignment technique that aligns feature maps at several sizes to create a multi-resolution feature map for enhanced feature fusion and model performance at many scales.

The fusion process is shown in Figure 3. Multiple convolution and pooling operations are performed on the input image to obtain feature maps of different sizes, and the last three layers are selected to construct the WCFPN, as shown in Equations (1) and (2).
(1)$${\vec{P}}^{\mathrm{in}}=\left(p_{l1}^{\mathrm{in}},p_{l2}^{\mathrm{in}},\ldots \right)$$
(2)$${\vec{P}}^{\mathrm{out}}=f\left({\vec{P}}^{\mathrm{in}}\right)$$

The multi-scale feature fusion in Equation (1) takes the feature maps from different resolutions and weights them, resulting in a richer and more comprehensive feature representation. Equation (2), on the other hand, cascades feature maps from different resolutions, thus combining feature information from different resolutions in the spatial dimension.

The problem of incomplete and inconsistent feature information is solved by the multi-scale feature fusion method, which can identify and locate targets more accurately. The research feature fusion method uses the fast normalization method, which can adjust the weights of different features by applying a normalization operation on each input feature to fuse feature maps at different scales more effectively, with the formula shown in Equation (3).
(3)$$O=\sum_{i}\left(\omega_{i}\cdot X_{i}\right)/\left(\epsilon +\sum_{j}\omega_{j}\right)$$
where O denotes the output feature mapping; $\omega_{i}$ is the weighting factor; $X_{i}$ denotes the input feature mapping, taken as 0.001.

Take $D_{2}$ layer output as an example. First, the $A_{1}$ layer is convolved 1 × 1 to obtain the feature mapping $B_{1}$. Second, the $B_{1}$ layer is fused with the $A_{2}$ layer to obtain the $B_{2}$ layer, and then the $B_{1}$ layer and $B_{2}$ layer are fused. Then, fuse both layers to obtain the intermediate layer mapping C. Finally, this layer is fed to $D_{2}$ for cross-scale connection and weighting to obtain the $D_{2}$ layer. The layer output can be expressed as Equations (4) and (5).
(4)$$D_{2}^{\mathrm{in}}=\mathrm{Conv}\left(\frac{\omega_{1}\cdot B_{2}^{\mathrm{in}}+\omega_{2}\cdot \mathrm{Res}\left(B_{1}^{\mathrm{in}}\right)}{W_{1}+W_{2}+\varepsilon }\right)$$
(5)$$D_{2}^{\mathrm{out}}=\mathrm{Conv}\left[\left(\omega_{1}^{\prime }\cdot B_{2}^{\mathrm{in}}+\omega_{2}^{\prime }\cdot B_{2}^{\mathrm{td}}+\omega_{3}^{\prime }\cdot \mathrm{Res}\left(B_{3}^{\mathrm{out}}\right)\right)/\omega_{1}^{\prime }+\omega_{2}^{\prime }+\omega_{3}^{\prime }+\varepsilon \right]$$
where $D_{2}^{\mathrm{in}}$ is the middle feature of the top-down path layer 2, and $D_{2}^{\mathrm{out}}$ is the output feature of the bottom-up path layer 2. Res denotes the up-sampling or down-sampling adjustment size, which is used to resize the feature map for feature fusion. All other features are constructed in the same way.

### 3.2. RWNMS

In target detection, sliding window or region extraction methods are usually used to generate candidate predictor frames, but the same target may be covered by multiple predictor frames, resulting in duplicate and inaccurate target detection results. The role of NMS operation is to select the predictor frame with the highest confidence among these overlapping predictor frames and remove other low-confidence results, and the calculation process is shown in Equation (6).
(6)$$D_{2}^{\mathrm{out}}=\mathrm{Conv}\left[\left(\omega_{1}^{\prime }\cdot B_{2}^{\mathrm{in}}+\omega_{2}^{\prime }\cdot B_{2}^{\mathrm{td}}+\omega_{3}^{\prime }\cdot \mathrm{Res}\left(B_{3}^{\mathrm{out}}\right)\right)/\omega_{1}^{\prime }+\omega_{2}^{\prime }+\omega_{3}^{\prime }+\varepsilon \right]$$

Although NMS can remove overlapping prediction frames, there may be a large amount of overlap in the prediction frames generated on multiple grids or multiple scales of feature maps, which will directly lead to the inability of the NMS algorithm to remove all the redundant prediction frames. To avoid the problem of redundant prediction frames, the study introduces the EIOU loss function, which considers not only the overlap between detection frames but also the similarity and confidence between them, etc.; by giving various detection findings distinct weights, the weighting approach may be utilized to pick and optimize outcomes or instance; while utilizing NMS, some significant targets or detection findings with higher confidence might be assigned larger weights and are hence more likely to be chosen. Additionally, a smoothing strategy is used to further increase the accuracy and robustness of detection by reducing issues like noise and oscillation by weighting the detection results of adjacent frames and averaging them when processing adjacent prediction frames, leading to more accurate and stable detection results; this can successfully boost detection’s resilience and generalization capabilities, better enabling it to handle detection jobs in complicated scenarios. The process of the improved RWI mechanism is shown in Figure 4, and its confidence score is calculated as Equation (7).
(7)$$c=p_{0}\times L_{\mathrm{EIOU}}.$$
where $L_{\mathrm{EIOU}}$ denotes the intersection ratio of the prediction frame and the true frame obtained based on the EIOU loss function, which can be expressed by Equation (8); $p_{0}$ denotes the probability of the presence of the target in the prediction frame, and if it exists then $p_{0}=1$, and vice versa $p_{0}=0$.
(8)$$L_{\mathrm{EIOU}}=\overset{s^{2}}{\underset{i=0}{\sum }}\overset{B}{\underset{j=0}{\sum }}\left(L_{\mathrm{IOU}}+L_{\mathrm{dis}}+L_{\mathrm{asp}}\right)\;L_{\mathrm{dis}}=\frac{\rho^{2}\left(b,b^{\mathrm{gt}}\right)}{{\left(w^{c}\right)}^{2}+{\left(h^{c}\right)}^{2}}\;L_{\mathrm{asp}}=\frac{\rho^{2}\left(w,w^{\mathrm{gt}}\right)}{{\left(w^{c}\right)}^{2}}+\frac{\rho^{2}\left(h,h^{\mathrm{gt}}\right)}{{\left(h^{c}\right)}^{2}}$$where $L_{\mathrm{dis}}$ denotes distance loss, $L_{\mathrm{asp}}$ denotes the phase loss, and $\rho^{2}\left(b,b^{\mathrm{gt}}\right)$ denotes the center point of the two frames b and $b^{\mathrm{gt}}$, $w^{c}$ and $h^{c}$ that denote the length and width of the smallest frame containing A and B, respectively, and the specific parameters are shown in Figure 5.

When there are targets of different scales or different shapes in the target detection, the traditional IOU calculation method is somewhat inaccurate because it only takes into account the ratio of the intersection and the concatenation of the detection frame and true labeled frame, ignoring the shape or size of the frame. To address this issue, EIOU introduces the aspect ratio and the Euclidean distance factor, increasing the variability between IOU values so that larger IOU values are closer to 1 and smaller IOU values are closer to 0, improving the accuracy and robustness of target detection.

In the calculation process, all prediction frames are first arranged in descending order according to the confidence score. Then, starting from the frame with the highest confidence score, all prediction frames are given weights and traversed again, the width and height of the current best frames are updated according to Equation (9), and the current best frame is finally obtained as the result after RWNMS.
(9)$${\mathrm{Box}}_{i=\max}\left[x_{1},y_{1},x_{2},y_{2}\right]=\frac{\sum_{i=1}^{\mathrm{target}}{\mathrm{cls}}_{i}\left[x_{1\_i},y_{1\_i},x_{2\_i},y_{2\_i}\right]}{\sum_{i=1}^{\mathrm{target}}{\mathrm{cls}}_{i}}$$
where ${\mathrm{box}}_{i=\max}[]$ denotes the box with the highest confidence level, and $\overset{\mathrm{target}}{\underset{i=1}{\sum }}$ denotes the prediction box from i = 1 to the target obtained by filtering. After that, the prediction box position is updated by Equation (9) to improve the stability of the prediction results.

## 4. Experimental Studies

### 4.1. Data Processing

To construct the data set required for this study, multiple segments of tunnel defective radar data were collected and post-processing operations such as image enhancement and image annotation were performed on these data. The final nearly 5700 images were obtained by LabelImg software labeling to form the data set of this study, and the defect images were classified into five categories such as BM, TK, KD, CS, and YBM, which denote five types of defects, namely, uncompacted, emptying, hollow, water-filled and severely uncompacted, respectively. Table 1 shows the distribution of the five types of defective images in the data set and Figure 6 shows some typical examples of defects.

### 4.2. Experimental Procedure

#### 4.2.1. Experimental Configuration

Radar detection equipment that emits radar waves and then receives the signals bounced back to obtain information about the parameters shown in Table 2.

The deep learning simulation experiments were built on a Linux system, using Python and PyTorch to build the deep learning framework. The hardware setup shown in Table 3 includes components such as CPU, GPU, memory, and storage.

#### 4.2.2. Evaluation Indicators

The experiments in this study use recall, mAP to evaluate the effectiveness of the model in detecting tunnel defects, which can be defined as Equation (10):
(10)$$\mathrm{Precision}=\frac{\mathrm{TP}}{\mathrm{TP}+\mathrm{FP}}\;\mathrm{Recall}=\frac{\mathrm{TP}}{\mathrm{TP}+\mathrm{FN}}\;\mathrm{AP}=\int_{0}^{1}P€\mathrm{dR}\;\mathrm{mAP}=\frac{\sum_{i=1}^{N}{\mathrm{AP}}_{i}}{N}\;{\mathrm{mAP}}_{\left[@0.5,@0.95\right]}=\left({\mathrm{mAP}}_{@0.5}+\cdots +{\mathrm{mAP}}_{@0.95}\right)/10$$
where TP represents the number of positive examples correctly classified, TN represents the number of negative examples correctly classified, FP represents the number of positive examples incorrectly classified, and FN represents the number of negative examples incorrectly classified.

## 5. Results

The study used the MFF-YOLO model for tunnel lining defect detection and successfully detected a wide range of defect types and sizes, with some of the results shown in Figure 7.

The loss function curve in Figure 8 was generated using a scatter plot to represent the training data. It is evident from observation that the MFF-YOLO model converges significantly more quickly than other models do, and in the 20th round, all loss functions begin to converge and subsequently begin to stabilize. While the convergence and stabilization of the loss function value suggest that the model has achieved the ideal state, the reduction in the loss function value shows that the model is gradually being optimized during the training phase.

In order to test the effectiveness of each model for each type of defect detection to highlight the importance of each module, the study tallied the accuracy results for different types as shown in Figure 9 and Table 4. 5 sections in Figure 9 represent each of the five types of defects, with different modules added to each section from left to right, where the leftmost is the original model and the rightmost red is the improved MFF-YOLO model. The results show that the improved model proposed in this paper outperforms the original model for all tunnel defect types, which further validates the effectiveness of the model in tunnel defect detection.

To more accurately assess the performance of the MFF-YOLO model proposed in this paper, the study conducted ablation tests and obtained the experimental results shown in Figure 10 and Table 5. The results show that the model MFF-YOLO achieves an accuracy of 89.4%. In addition, through the ablation tests, we also verified that the model has a strong robustness and generalization capability for effective tunnel defect detection under different environments and conditions.

The research in Table 5 shows that after adding WCFPN, the mAP, Precision, Recall and Gflops of the model are significantly improved through deep fusion extraction of feature information. After adding RWNMS to reprocess the prediction box, although the accuracy improvement is not as obvious as WCFPN, it keeps the FPS stable. When the two modules are combined to construct the MFF-YOLO model in this study, the result is a synergistic impact that outperforms the effect of each module acting alone and raises the model’s total performance ceiling.

Figure 11 presents a visualization of the test results of this study on the data set, allowing for a more intuitive understanding of the performance and performance of the model, and thus better model improvement and optimization.

These results show that the improved algorithm proposed in this study has high performance and practicality in tunnel defect detection and can be better adapted to a variety of different types and sizes of defect detection tasks.

## 6. Conclusions

In this study, we developed an MFF-YOLO model based on multi-scale feature fusion technology. In the neck network, WCFPN is designed based on a weighted fusion strategy to improve the ability to obtain feature information from multiple dimensions. At the predicted end, the prediction mechanism of RWNMS is designed in combination with a weighted smoothing strategy. Finally, the EIOU loss function is improved by considering the aspect ratio and Euclidean distance.

The feasibility of the strategy was evaluated by testing the tunnel defect data set. The result shows that the MFF-YOLO model’s recall rate and accuracy rates were 7.1% and 6.0% higher than those of the YOLOv5 model, reaching 89.5% and 89.4%, respectively.

Compared to conventional image processing approaches, deep learning-based tunnel defect detection algorithms may learn higher-level feature representations from a huge quantity of data, improving the identification and classification of flaws. Additionally, automated and intelligent deep learning-based detection strategies can increase the effectiveness and dependability of detection. In the future, we will continue to explore ways to improve model defect detection performance, including reducing model complexity, improving detection accuracy, and optimizing data set quality.

## Figures and Tables

**Figure 1 sensors-23-06490-f001:**
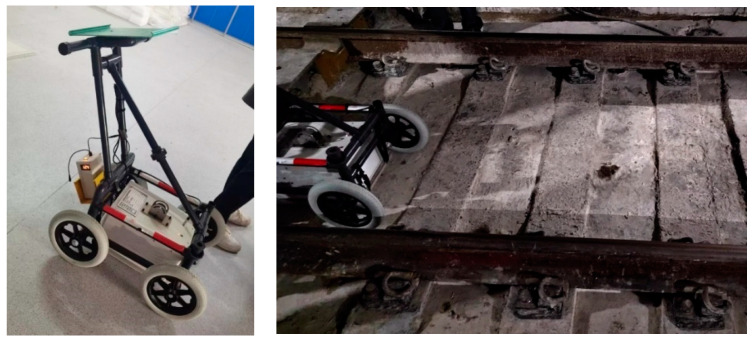
Ground-Penetrating Radar Vehicle TGRI-GPR.

**Figure 2 sensors-23-06490-f002:**
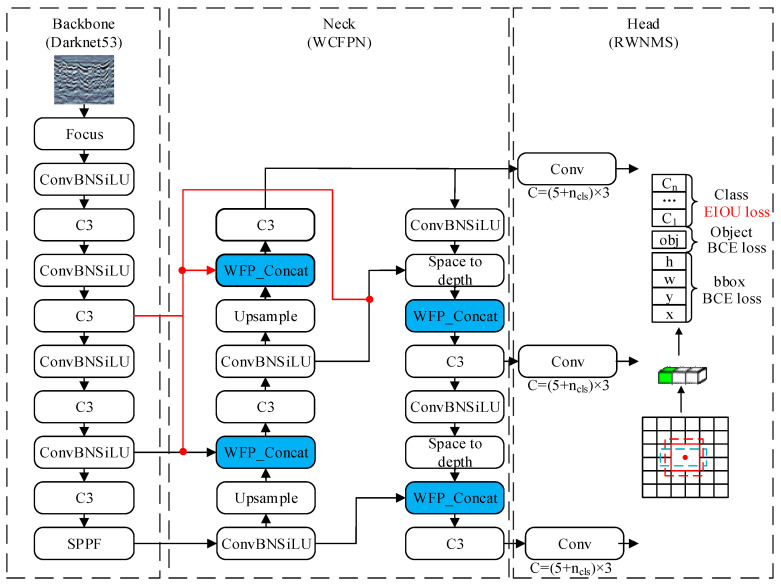
Structure of MFF-YOLO. Where the black solid lines indicate the input-output paths and the red solid lines are the additional input-output paths of the improved model, and the black solid lines are used to form a grid map in the detection head part, and the red and blue dashed boxes indicate the location of the prediction box.

**Figure 3 sensors-23-06490-f003:**
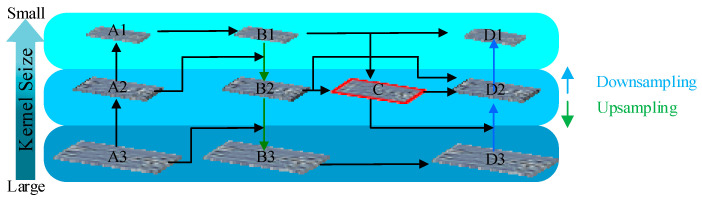
Structure of WCFPN. Where $A_{1}$, $A_{2}$, …, $D_{1}$, $D_{2}$, $D_{3}$ represent different upper and lower sampling layers respectively.

**Figure 4 sensors-23-06490-f004:**
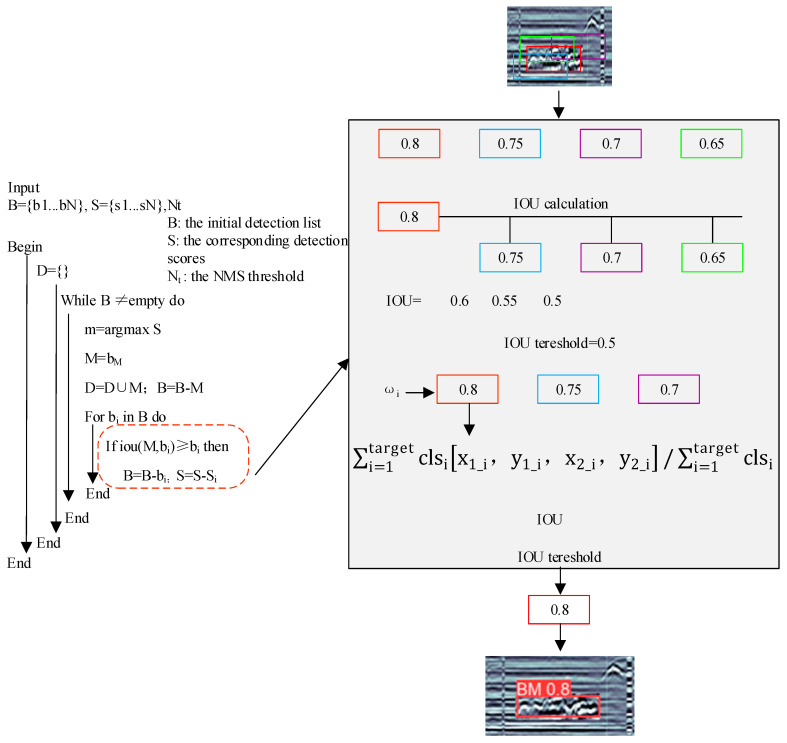
Improved RWI module. Where different colors represent different prediction frames.

**Figure 5 sensors-23-06490-f005:**
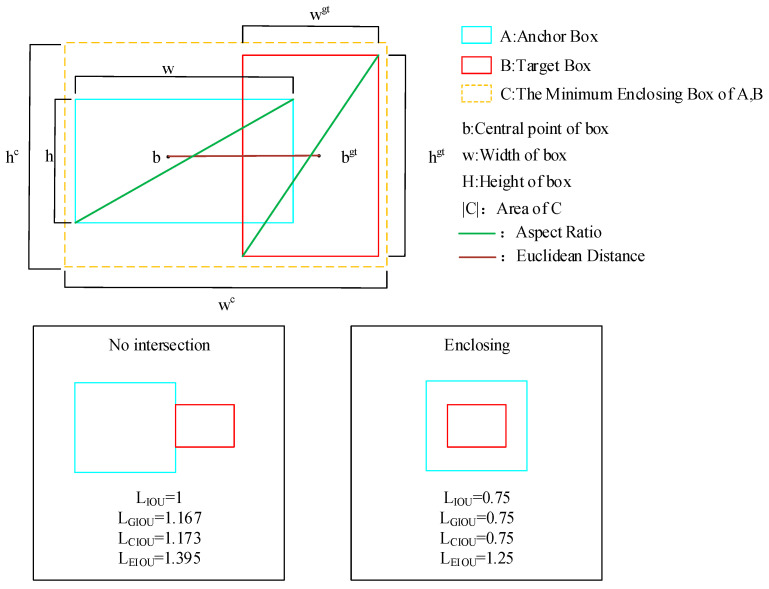
Some parameters of EIOU loss function.

**Figure 6 sensors-23-06490-f006:**
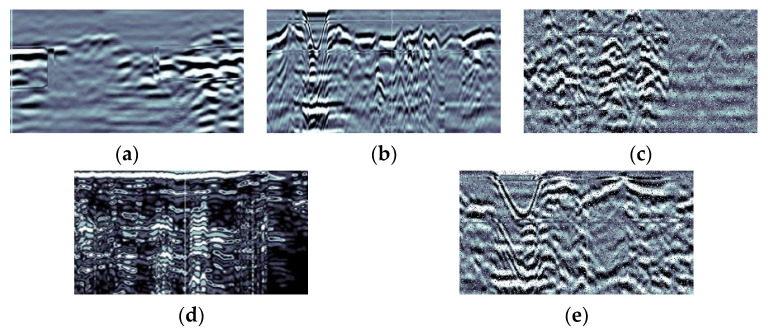
Defect sample example. (**a**) uncompacted, (**b**) emptying, (**c**) hollow, (**d**) water-filled, (**e**) severely uncompacted.

**Figure 7 sensors-23-06490-f007:**
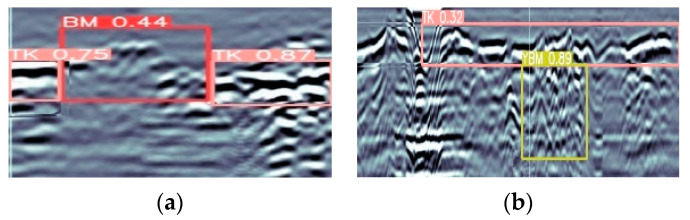
Detection of different categories of defects. (**a**) Effect of BM detection, (**b**) Effect of TK detection, (**c**) Effect of KD detection, (**d**) Effect of CS detection, (**e**) Effect of YBM detection.

**Figure 8 sensors-23-06490-f008:**
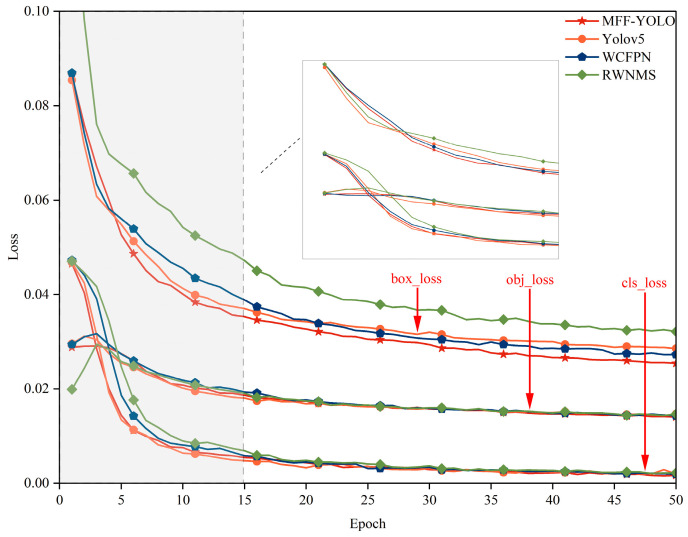
MFF-YOLO loss curve.

**Figure 9 sensors-23-06490-f009:**
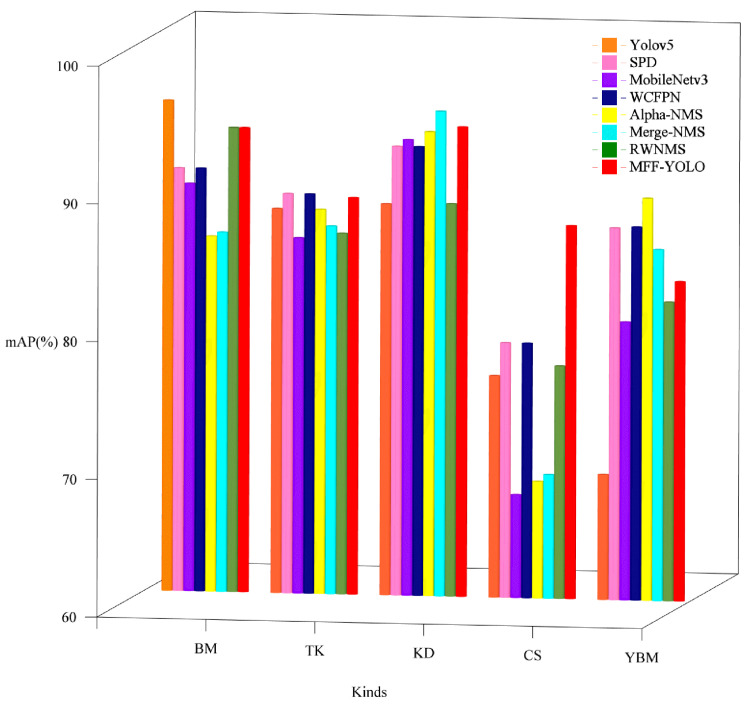
Comparison of detection effects of different models for different categories.

**Figure 10 sensors-23-06490-f010:**
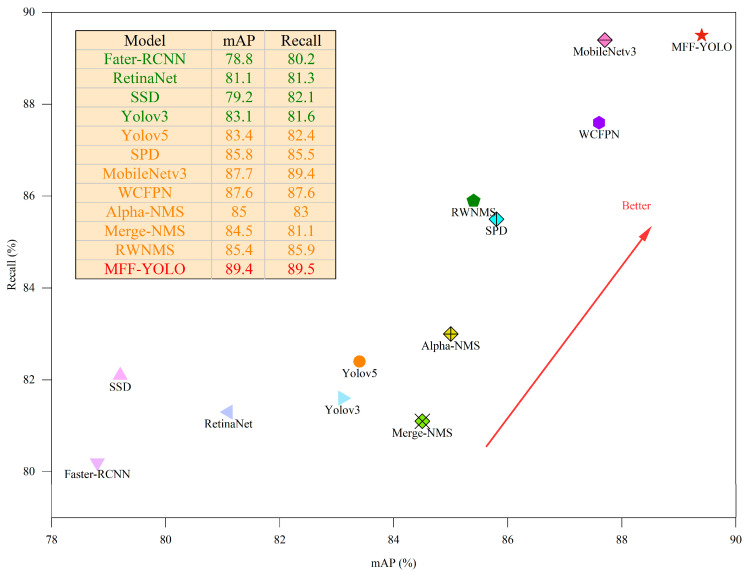
Compare of mAP and Recall.

**Figure 11 sensors-23-06490-f011:**
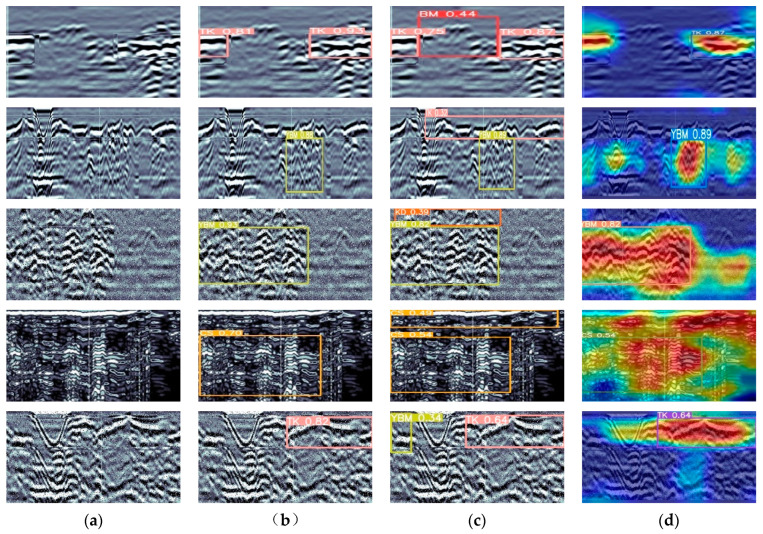
Detection result plots. Where (**a**) is the original defect maps, (**b**) is the original model detection result maps, (**c**) is the improved MFF-YOLO model detection result maps, and (**d**) is the result visualization maps.

**Table 1 sensors-23-06490-t001:** Distribution of tunnel defect data sets.

Defect Type	Training Set	Validation Set	Test Set	Label
BM	1301	114	76	0
TK	1815	127	139	1
KD	904	78	254	2
CS	1136	80	197	3
YBM	1175	111	107	4

**Table 2 sensors-23-06490-t002:** Radar rover-related parameters.

Name	Configure
Equipment Model	TGRI-GPR
Center Frequency	200 MHz
Operating Bandwidth	100–500 MHz
Depth of detection	3 m
Dynamic Range	40 dB

**Table 3 sensors-23-06490-t003:** Experimental hardware configuration.

Name	Configure
Operating System	Linux
Video Card	NVIDIA RTX3090
Video Memory	24 G
Processor	Intel€ Core i3-8100
Programming Language	Python
Deep training framework	PyTorch
Programming Platforms	Pycharm

**Table 4 sensors-23-06490-t004:** Comparison of experimental results of each model in different defects.

	**Type**	**Average Accuracy Rate (%)**				
Model		BM	TK	KD	CS	YBM
Yolov5		95.6	87.9	88.4	76.1	69.1
SPD		90.7	89.0	92.6	78.5	87.0
MobileNetv3		89.6	85.8	93.1	67.5	80.2
WCFPN		90.7	89.0	92.6	78.5	87.1
Alpha-NMS		85.8	87.9	93.7	68.5	89.2
Merge-NMS		86.1	86.7	95.2	69.0	85.5
RWNMS		93.7	86.2	88.5	76.9	81.7
MFF-YOLO		93.7	88.8	94.1	87.1	83.2

**Table 5 sensors-23-06490-t005:** Results of ablation experiments.

Name	mAP@0.5 (%)	mAP@0.5:0.95 (%)	Precision (%)	Recall (%)	Gflops	FPS
Fater-RCNN	78.8	48.8	78.6	80.2	88.2	18.2
RetinaNet	81.1	49.8	79.7	81.3	70.3	18.3
SSD	79.2	47.0	75.2	82.1	15.2	65.8
Yolov3	83.1	49.1	80.3	81.6	154.6	16.4
Yolov5	83.4	47.9	75.4	82.4	16.8	66.6
SPD	85.8	49.1	80.3	85.5	16.0	71.4
MobileNetv3	87.7	50.4	81.0	89.4	15.3	50.0
WCFPN	87.6	51.6	82.2	87.6	33.1	52.6
Alpha-NMS	85.0	51.3	78.6	83.0	15.8	71.4
Merge-NMS	84.5	50.5	78.3	81.1	15.8	71.4
RWNMS	85.4	49.7	80.8	85.9	16.8	66.6
MFF-YOLO	89.4	50.8	82.2	89.5	33.3	50.0

## Data Availability

The data used to support the findings of this study are included within the article.

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
