# Peer review of "MFF-YOLO: An Accurate Model for Detecting Tunnel Defects Based on Multi-Scale Feature Fusion"

_sensors, 2023, doi:10.3390/s23146490_

Round 1

Reviewer 1 Report

The manuscript titled "MFF-YOLO: An Accurate Model for Detecting Tunnel Defects Based on Multi-scale Feature Fusion" presents a technique for the detection of tunnel lining defects using a convolutional neural network. The manuscript highlights the importance of routine inspection for tunnel linings and addresses the challenges associated with detecting defects in complex tunnel environments. The proposed MFF-YOLO technique incorporates multi-scale feature fusion, a re-weighting screening approach, and an improved loss function to enhance the model's effectiveness. The experimental results demonstrate improvements in recall, accuracy, and the ability to identify previously disregarded or misdetected targets compared to the YOLOv5 model. Overall, the manuscript is well-structured and provides clear explanations of the proposed approach and its impact on tunnel lining detection performance.

The introduction effectively emphasizes the significance of routine inspection for tunnel linings and the challenges associated with defect detection in tunnel environments. However, consider providing more specific information about the application of deep learning techniques in object detection. Some relevant examples are: DOI:10.3390/s22093467, DOI: 10.1007/s00170-022-10335-8.

The manuscript describes the proposed MFF-YOLO technique and its components, such as multi-scale feature fusion, re-weighting screening approach, and improved loss function. While these components are mentioned, it would be beneficial to provide more details on their individual functionalities and how they contribute to the overall effectiveness of the model.

The manuscript mentions that the MFF-YOLO model is able to identify targets that were previously disregarded or misdetected by the previous model. It would be valuable to provide specific examples or qualitative analysis of these improved detections to demonstrate the effectiveness of the proposed approach.

In the conclusion, highlight the potential benefits, challenges, and any future research directions or improvements that could be explored to further enhance the detection performance.

The language and style of the manuscript are generally clear and concise, allowing readers to understand the proposed approach and experimental results effectively.

Author Response

Dear Reviewer,

We appreciate you reading our paper, MFF-YOLO: An Accurate Model for Detecting Tunnel Defects Based on Multi-scale Feature Fusion, and providing your valuable comments. We appreciate your comments, and the attached Response to Reviewer 1 Comment has comprehensive answers to all of your comments. Every comment and suggestion is taken seriously, and we have taken steps to refine and improve them. We really hope that you will accept and appreciate these steps.

We are more than eager to accept your suggestions for improvement and will keep working hard to make our work better if you feel there is anything that is improper or could be done better. We will continue to conduct our research with a professional attitude and degree of rigor, as we always have, and we will increase our contributions to the academic community while also raising the bar on our level and quality of research.

Once again, we want to thank you for your review remarks as well as for your encouragement and support of our study.

Thank you and best regards.

Yours sincerely,

Anfu Zhu, Bin Wang, Jiaxiao Xie, Congxiao Ma

Reviewer 2 Report

The monitoring of works and safety features in sinkhole structures is an important and at the same time difficult task. The use of near-sensing methods is commonly used in these tasks The authors present a solution based on artificial intelligence (AI) using a convolutional neural network (CNN).  An interesting application of YOLO technology for the detection of tunnel lining defects. Publication prepared with care and I find no weaknesses. 

Author Response

Dear reviewer,

We sincerely appreciate you taking the time to read and comment on our publication, MFF-YOLO: An Accurate Model for Detecting Tunnel Defects Based On Multi-scale Feature Fusion. Knowing that you think highly of our work gives us a lot of comfort and serves as a powerful motivator for us. We are grateful for your acknowledgement and will keep striving to raise the bar for our research's level and quality.

We'll make sure that our study meets higher academic standards and advances the field.

We want to once again thank you for your help and advice. We shall make every effort to further our research.

Thank you and best regards.

Yours sincerely,

Anfu Zhu, Bin Wang, Jiaxiao Xie, Congxiao Ma

Reviewer 3 Report

The detection of tunnel lining defects is still fraught with difficulties because of the complexity of the tunnel environment. The submitted manuscript proposed an accurate model for detecting tunnel defects based on multi-scale feature fusion. At present, most existing studies focusing on improving the YOLO models have been published (Damage Detection and Localization of Bridge Deck Pavement Based on Deep Learning, Toward High-Precision Crack Detection in Concrete Bridges Using Deep Learning). The motivation of proposing a new model and the differences between the proposed model needs to be explained in detail. In addition, the following several comments are suggested to be considered before the manuscript can be accepted.

1. There are some grammar mistakes in the manuscript, for example, in line 16, “misdetected” should be changed to “miss detection”, in line 103, “ormaliza” should be replaced by “normalized”. The manuscript is suggested to be polished by native speaker to improve its readability.

2. The verification on the VOC dataset should be placed before the detection of tunnel defects, and figures of the results are blurred. In addition, is this step unnecessary? The authors are suggested to analyze the performance of the network immediately after the network is improved.

3. Some of the AP values in Fig.7 and Fig.11 are not high, and it seems that they do not reach the high precision mentioned in the conclusion. Explanations are suggested to be included in that part.

4. Abstract and conclusion of the manuscript may need to be r optimized to reflect a clear explanation of the primary contributions this research makes to the body of knowledge. And the paper could be shortened without compromising its message.

1. There are some grammar mistakes in the manuscript, for example, in line 16, “misdetected” should be changed to “miss detection”, in line 103, “ormaliza” should be replaced by “normalized”. The manuscript is suggested to be polished by native speaker to improve its readability.

Author Response

Dear Reviewer,

We appreciate you reading our paper, MFF-YOLO: An Accurate Model for Detecting Tunnel Defects Based on Multi-scale Feature Fusion, and providing your valuable comments. We appreciate your comments, and the attached Response to Reviewer 3 Comment has comprehensive answers to all of your comments. Every comment and suggestion is taken seriously, and we have taken steps to refine and improve them. We really hope that you will accept and appreciate these steps.

We are more than eager to accept your suggestions for improvement and will keep working hard to make our work better if you feel there is anything that is improper or could be done better. We will continue to conduct our research with a professional attitude and degree of rigor, as we always have, and we will increase our contributions to the academic community while also raising the bar on our level and quality of research.

Once again, we want to thank you for your review remarks as well as for your encouragement and support of our study.

Thank you and best regards.

Yours sincerely,

Anfu Zhu, Bin Wang, Jiaxiao Xie, Congxiao Ma

Round 2

Reviewer 3 Report

1. The reviewer think it is not appropriate to use the voc data to detect the cats and other animals to validate the performance of MFF-YOLO for detecting the tunnel defects. More tunnel images are required to validate the accuracy and robustness of the improved model.

2. references about using machine learning algorithms to detecting strcutural cracks and crossions are also suggested to be included in the introduction, e.g., Damage Detection and Localization of Bridge Deck Pavement Based on Deep Learning, Toward High-Precision Crack Detection in Concrete Bridges Using Deep Learning, etc. 

1. The reviewrer think it is not appropriate to use the voc data to detect the cats and other animals to validate the performance of MFF-YOLO for detecting the tunnel defects. More tunnel images are required to validate the accuracy and robustness of the improved model.

Author Response

Dear Reviewer,
We appreciate you reading our paper, MFF-YOLO: An Accurate Model for Detecting Tunnel Defects Based on Multi-scale Feature Fusion, and providing your valuable comments.   We appreciate your comments, and the attached Response to Reviewer Comment has comprehensive answers to all of your comments.   Every comment and suggestion is taken seriously, and we have taken steps to refine and improve them.   We really hope that you will accept and appreciate these steps.
We are more than eager to accept your suggestions for improvement and will keep working hard to make our work better if you feel there is anything that is improper or could be done better.   We will continue to conduct our research with a professional attitude and degree of rigor, as we always have, and we will increase our contributions to the academic community while also raising the bar on our level and quality of research.
Once again, we want to thank you for your review remarks as well as for your encouragement and support of our study.

Thank you and best regards.
Yours sincerely,
Anfu Zhu, Bin Wang, Jiaxiao Xie, Congxiao Ma
